# A serum proteome signature to predict mortality in severe COVID-19 patients

Franziska Völlmy[1,2], Henk van den Toorn[1,2], Riccardo Zenezini Chiozzi[1,2], Ottavio Zucchetti[3],
Alberto Papi[4], Carlo Alberto Volta[5], Luisa Marracino[6], Francesco Vieceli Dalla Sega[7], Francesca Fortini[7],
Vadim Demichev[8,9,10], Pinkus Tober-Lau[11] ⓘ, Gianluca Campo[3,7], Marco Contoli[4], Markus Ralser[8,9],
Florian Kurth[11,12], Savino Spadaro[5] ⓘ, Paola Rizzo[6,7], Albert JR Heck[1,2] ⓘ

Here, we recorded serum proteome profiles of 33 severe COVID-19 patients admitted to respiratory and intensive care units because of respiratory failure. We received, for most patients, blood samples just after admission and at two more later time points. With the aim to predict treatment outcome, we focused on serum proteins different in abundance between the group of survivors and non-survivors. We observed that a small panel of about a dozen proteins were significantly different in abundance between these two groups. The four structurally and functionally related type-3 cystatins AHSG, FETUB, histidine-rich glycoprotein, and KNG1 were all more abundant in the survivors. The family of inter-$\alpha$-trypsin inhibitors, ITIH1, ITIH2, ITIH3, and ITIH4, were all found to be differentially abundant in between survivors and non-survivors, whereby ITIH1 and ITIH2 were more abundant in the survivor group and ITIH3 and ITIH4 more abundant in the non-survivors. ITIH1/ITIH2 and ITIH3/ITIH4 also showed opposite trends in protein abundance during disease progression. We defined an optimal panel of nine proteins for mortality risk assessment. The prediction power of this mortality risk panel was evaluated against two recent COVID-19 serum proteomics studies on independent cohorts measured in other laboratories in different countries and observed to perform very well in predicting mortality also in these cohorts. This panel may not be unique for COVID-19 as some of the proteins in the panel have previously been annotated as mortality markers in aging and in other diseases caused by different pathogens, including bacteria.

## Introduction

The coronavirus disease 2019 (COVID-19) pandemic caused by severe acute respiratory syndrome coronavirus 2 (SARS-CoV-2) has affected many people with a worrying fatality rate up to 60% for critical cases. Not all people infected by the virus are affected equally. Several parameters have been defined that may influence and/or predict disease severity and mortality, with age, gender, body mass, and underlying comorbidities being some of the most well established. To delineate best treatments and recognize disease severity early on, so that clinicians can decide on treatment options, it would be very helpful to discover markers helping to define disease severity, ideally having prognostic value, and/or predict a specific phase of the disease (1 Preprint). Unfortunately, not many prognostic biomarkers are yet available that can distinguish patients requiring immediate medical attention and estimate their associated mortality rates.

Here, we attempted to contribute to this urgent need aiming to find serum biomarkers that can be used to predict mortality in a group of COVID-19 patients. For the present purpose, we prospectively assessed serum protein levels at different time points by using mass spectrometry–based serum proteomics in a cohort of moderate-to-severe COVID-19 patients admitted to hospital because of respiratory failure (ATTAC-Co study–registered at www.clinicaltrials.gov number NCT04343053).

Given the central role of proteins in biological processes as a whole, and in particular in diseases, we applied mass spectrometry–based proteomics to identify protein biomarkers that could discriminate between the COVID-19 patients who recovered and those who did not

[1]Biomolecular Mass Spectrometry and Proteomics, Bijvoet Center for Biomolecular Research and Utrecht Institute for Pharmaceutical Sciences, University of Utrecht, Utrecht, The Netherlands   [2]Netherlands Proteomics Center, Utrecht, The Netherlands   [3]Cardiology Unit, Azienda Ospedaliero-Universitaria di Ferrara, University of Ferrara, Ferrara, Italy   [4]Respiratory Section, Department of Translational Medicine, University of Ferrara, Ferrara, Italy and Respiratory Disease Unit, Azienda Ospedaliero-Universitaria di Ferrara, Italy   [5]Department of Translational Medicine University of Ferrara, Ferrara, Italy and Intensive Care Unit, Azienda Ospedaliero-Universitaria di Ferrara, Italy   [6]Department of Translational Medicine and Laboratory for Technology of Advanced Therapies (LTTA), University of Ferrara, Ferrara, Italy   [7]Maria Cecilia Hospital, GVM Care & Research, Cotignola, Italy   [8]Charité–Universitätsmedizin Berlin, Department of Biochemistry, Berlin, Germany   [9]The Francis Crick Institute, Molecular Biology of Metabolism Laboratory, London, UK   [10]The University of Cambridge, Department of Biochemistry and Cambridge Centre for Proteomics, Cambridge, UK   [11]Charité–Universitätsmedizin Berlin, Department of Infectious Diseases and Respiratory Medicine, Berlin, Germany   [12]National Phenome Centre and Imperial Clinical Phenotyping Centre, Department of Metabolism, Digestion and Reproduction, Imperial College London, London, UK

Correspondence: a.j.r.heck@uu.nl

survive. Several proteomic studies (2, 3, 4, 5) or even multi-omics studies, including proteomics (6, 7), have to date investigated the serum or plasma of COVID-19 patients for the most part comparing a cohort of COVID-19 patients with control subjects (no disease) (2, 3). Of special relevance to our work, an extensive study by Demichev et al (8) has already investigated the temporal aspect of COVID-19 progression in individuals to predict outcome and future disease progression. Although their cohort did comprise some patients who did not survive, for the most part, the subjects recovered and were discharged. The unique cohort described in our study allows us to focus on survivors compared with non-survivors and to define clinical classifiers predicting outcome by using subjects recovered from the disease as a control group.

We chose a robust data-independent acquisition (DIA) approach to profile the serum of this patient cohort, as this method circumvents the semi-stochastic sampling bias specific to standard shotgun proteomics, and benefits from high reproducibility. The DIA approach, although not novel (9, 10), has recently increased in popularity in part due to new hardware and software solutions (11) but also due to the efforts of the proteomics community to develop DIA setups that do not require a reference spectral library. In this work, we chose to exploit the DIA-NN software suite (12), which makes use of deep neural networks and signal correction to process the complex spectral maps that arise from DIA experiments. This results in a reduction of interfering spectra and in confident statistically significant identifications thanks to the use of neural networks to distinguish between target and decoy precursors.

We observed that the group of survivors and non-survivors could be well separated by just a small group of around a dozen abundant serum proteins, and pleasingly best at the first time point, that is, shortly after admission to the ICU. This panel of proteins includes various functionally related proteins, including all major serum type-3 cystatins (histidine-rich glycoprotein [HRG], fetuin-B [FETUB], fetuin-A [FETUA], and kininogen-1 [KNG1]) and several protease inhibitors (Serpin A2 [SERPINA2], inter-alpha-inhibitor heavy chain 1 and 2 [ITIH1 and ITIH2]). As this panel is already able to distinguish the patient groups at the early onset, it may have a good predictive value. Statistically most significant, the type-3 cystatin HRG and FETUB were consistently more abundant in survivors than in non-survivors. These two proteins have previously been identified as predictors of mortality in patients affected by *Staphylococcus aureus* bacteremia (13), but also as general mortality markers in studies looking at aging (14). Therefore, the panel we observe here may not be specific for patients suffering from COVID-19, and be more generally applicable to predict mortality risk (15).

Finally, the performance of the mortality risk panel defined here was evaluated against two recent COVID-19 serum proteomics studies on independent cohorts and observed to be able to predict mortality also in these cohorts, validating that plasma proteomics may produce reproducible and meaningful biomarker panels.

# Results

For our proteomics analysis, we analyzed 33 patients, selected to obtain two comparable groups in terms of sex (73% males), pharmacological treatment and comorbidities, as well as age as much as

was possible (median age 71 ± 7.6 versus 65 ± 9.8, survivor versus deceased) (Fig S1A and B and Tables S1 and S2). Of the 17 survivors, 14 had blood withdrawn at three time points and 3 patients at only two time points. In the non-survivor group of 16 patients, 6 patients had blood sampled at three time points, whereas 5 had two time points, and 5 had only one time point (Fig 1A). Following a serum proteomics sample preparation workflow optimized in our group and by others (16, 17), we set out to process all 81 samples simultaneously to avoid introducing batch effects which may confound the results. DIA was performed by analyzing all samples in a randomized order to also further avoid batch effects. The cohort and the experimental approach used are schematically summarized in Fig 1A–E and further described in the Materials and Methods section.

In total, and after excluding numerous detected variable immunoglobulin protein fragments (Table S3), we could quantify a mean number of 452 proteins per sample (min = 302, max = 578) (Fig 2B and C). In serum proteomics, it has been well established that the total intensity of a protein in label-free quantification (i.e., label-free quantification [LFQ]- or intensity based absolute quantification [IBAQ]-values) can be used as a proxy for protein concentrations. To better relate the abundance of serum proteins to clinical data, we converted the median log label-free quantified values per protein from our mass spectrometry experiments into serum protein concentrations. For this conversion, we performed a linear regression with 22 known reported average values of proteins in serum (A2M, B2M, C1R, C2, C6, C9, CFP, CP, F10, F12, F2, F7, F8, F9, HP, KLKB1, MB, MBL2, SERPINA1, TFRC, TTR, and VWF) (18). This analysis yielded a sensible correlation coefficient of $R^2 = 0.78$ between the proteomics concentration measurements and the average values reported in literature (Fig S2). We therefore decided to convert all mass spectrometric values by using this concentration scale (mg/dl) (Table S4). Moreover, gene names are used throughout this work as identifiers when referring to proteins we measured.

A first look at the proteins present in our serum samples across all patients revealed a few potential clear outliers. At single time points and in single patients, several proteins originating from either red blood cells (e.g., *HBA1*, *HBB*, *CA1*, *CA2*, and *PRDX2*) or fibrinogen (e.g., *FGA*, *FGB*, and *FGG*) were extremely prominent (Fig S3A and B). These features are more often observed in serum proteomics and are likely caused by sample preparation artefacts (19). Fortunately, they do not negatively affect the abundance measurements of the other serum proteins. Therefore, we decided to exclude a panel of well-described RBC contaminants (Table S5) from all further analyses. We then sought to determine the differences in serum proteome expression stratifying survivors and deceased patients. For this, at each time point separately, we performed a two-sample *t* test and identified proteins significantly differentially expressed between survivors and non-survivors (Fig 2A and Table S6). At t1, just after admission to the ICU, 42 proteins were found to be significantly differentially expressed (28 higher in survivors and 14 higher in non-survivors) taking as threshold a *P*-value of 0.05. At t2, 30 proteins were found to be differentially expressed (19 higher in survivors and 11 higher in non-survivors) (Fig S4A). Finally, at t3, 19 proteins were significantly different (10 higher in survivors and 9 higher in non-survivors) (Fig S4B). In our dataset, two proteins were significantly different between survivors and non-survivors at all time points (*HRG* and haptoglobin-related

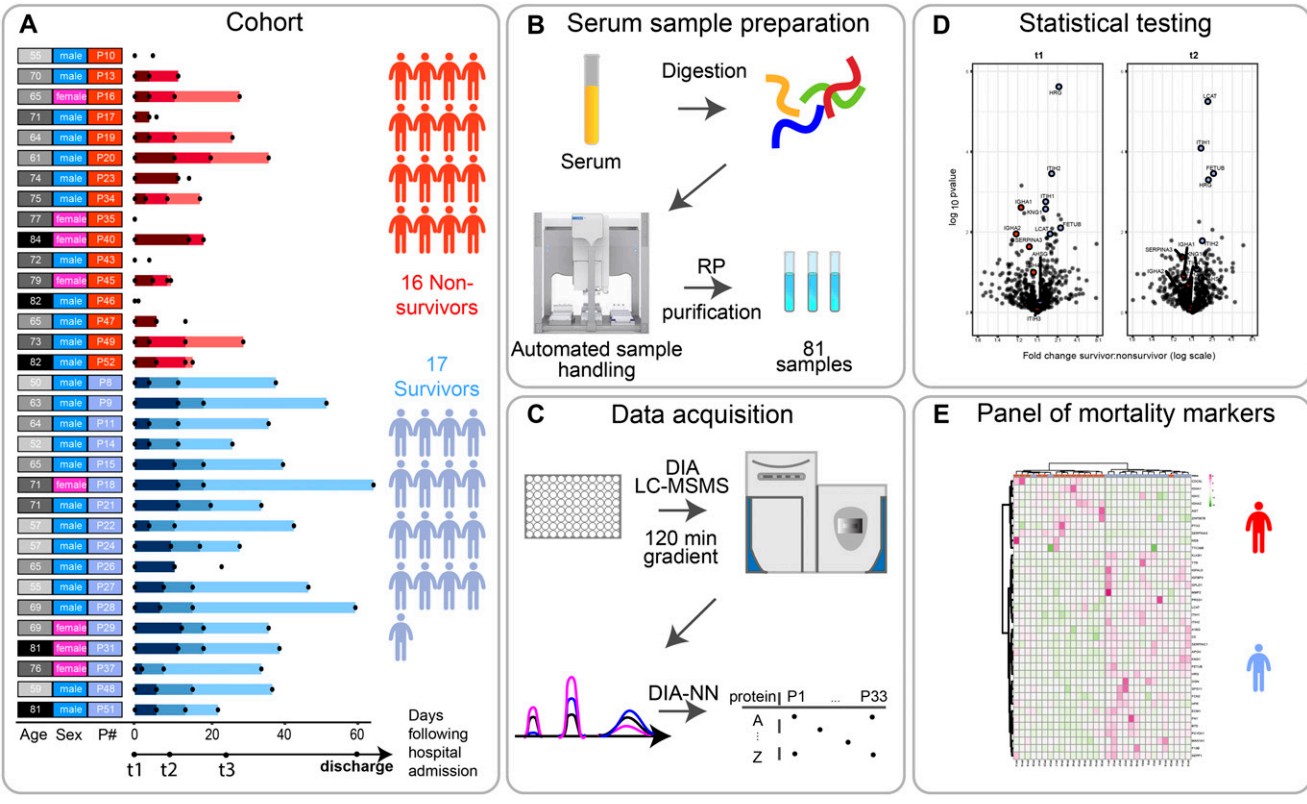

**Figure 1. Scheme of the cohort and timing of the blood sample collection, based on each patient's admission to the hospitalization.**
**(A)** Serum samples were collected from 33 individuals (17 survivors, 16 deceased) diagnosed with SARS-CoV-2 infection, at one, two, or three time points after their admission to the clinic (t0). The time points t1, t2, and t3 represent blood collections at 96 h, 14 d, and later than 14 d after they arrived at the ICU of the hospital. The date of discharge from the hospital is recorded here, although no blood was collected at that moment. The numbers represent the elapsed number of days starting from t1 for each patient. These are represented by color gradients ranging from dark to light the longer the duration of the stay in the clinic. Patients where no color timeline is represented, indicate cases for which no consecutive temporal collection points were available. Patient descriptors including age and gender as well as indexes used throughout this report are provided to the right of the timelines and color coded. Patient's age is binned (10 yr/bin) and the darker the greyscale in column 1 the older the patient. The gender of each patient is marked in column 2 as blue and pink. Patient indexes are color-coded for patient outcomes, with survivors in blue and deceased SARS-CoV-2 patients in red in column 3. **(B)** Serum samples were proteolytically digested and the resulting tryptic peptides were purified using reverse-phase cartridges on an autosampler robot. **(C)** The samples were analyzed by liquid chromatography tandem mass spectrometry (LC-MS/MS) applying a data-independent acquisition strategy. Spectra were extracted using DIA-NN yielding as a measure abundance for each protein. **(D)** Two-sided *t* tests were performed to determine significantly regulated proteins comparing survivors and non-survivors. **(E)** These differentially regulated proteins were found to be largely functionally related, and define a potential panel of mortality markers, by which we can stratify patients that might ultimately overcome or succumb from the infection, which can be diagnosed already at an early time point in the clinic.

protein [*HPR*]). Of note, at the latest time point, t3, we unfortunately could include only a lower number of non-survivors, likely affecting the observed *P*-values adversely. The overlap in proteins (denoted by their gene names throughout this work) significantly different between survivors and non-survivors at the first two time points was nine (*HRG* [*P*-value at t1: $1.88 \times 10^{-6}$; t2: $5.32 \times 10^{-4}$; t3: $3.54 \times 10^{-2}$], *FETUB* [*P*-value at t1: $6.9 \times 10^{-3}$; t2: $3.12 \times 10^{-4}$; t3: $2.53 \times 10^{-1}$], *ITIH1* [*P*-value at t1: $1.76 \times 10^{-3}$; t2: $8.08 \times 10^{-5}$; t3: $1.17 \times 10^{-1}$], *ITIH2* [*P*-value at t1: $2.94 \times 10^{-4}$; t2: $1.5 \times 10^{-2}$; t3: $2.75 \times 10^{-1}$], *HPR* [*P*-value at t1: $2.16 \times 10^{-2}$; t2: $3.67 \times 10^{-2}$; t3: $3.13 \times 10^{-3}$], *SERPINA3* [*P*-value at t1: $2.17 \times 10^{-2}$; t2: $3.61 \times 10^{-2}$; t3: $2.64 \times 10^{-1}$], *LCAT* [*P*-value at t1: $9.81 \times 10^{-3}$; t2: $4.85 \times 10^{-6}$; t3: $8.12 \times 10^{-2}$], *IGFALS* [*P*-value at t1: $9.87 \times 10^{-3}$; t2: $9.29 \times 10^{-3}$; t3: $5.11 \times 10^{-1}$], and *IGFBP3* [*P*-value at t1: $3.77 \times 10^{-3}$; t2: $1.02 \times 10^{-2}$; t3: $7.28 \times 10^{-1}$]). The nature of the first two time points as well as the number of patient samples available at these time points led us to focus first on these as they would enable the most appropriate comparison (at t1 there are 16 survivors and 15 non-survivors, compared to t2 with 15 versus 11 and t3 with 16 versus 6). We did choose to first focus on the

data for t1, as this represents also the most narrowly defined time-frame, when compared to t2 and t3. Although we thus do deliberately not focus on time point t3, due to the lower statistics, we did observe at this last time point that several neutrophil originating proteins, such as *MPO*, *PRTN3*, and *LCN2*, indicated in yellow in Fig 2A, were more abundant in the non-survivors. Four proteins that were clearly significant at time point 1 but that did not pass the significance threshold at time point 2, and are of particular interest were *FN1* (*P*-value at t1: $4.82 \times 10^{-3}$; t2: $1.61 \times 10^{-1}$; t3: $4.37 \times 10^{-1}$), *IGHA1* (*P*-value at t1: $2.45 \times 10^{-3}$; t2: $1.21 \times 10^{-1}$; t3: $2.67 \times 10^{-1}$), *IGHA2* (*P*-value at t1: $1.11 \times 10^{-2}$; t2: $7.9 \times 10^{-1}$; t3: $1.25 \times 10^{-1}$), and *KNG1* (*P*-value at t1: $2.67 \times 10^{-3}$; t2: $8.29 \times 10^{-2}$; t3: $5.68 \times 10^{-1}$).

With a focus on the proteins showing differences in abundance at time point 1, we applied an unsupervised clustering approach and found as expected that the differentially regulated proteins that resulted from the analysis of time point 1 samples show a clear cluster that is distinct between the survivors and some non-survivors (Fig 3A). These same differentially regulated proteins also

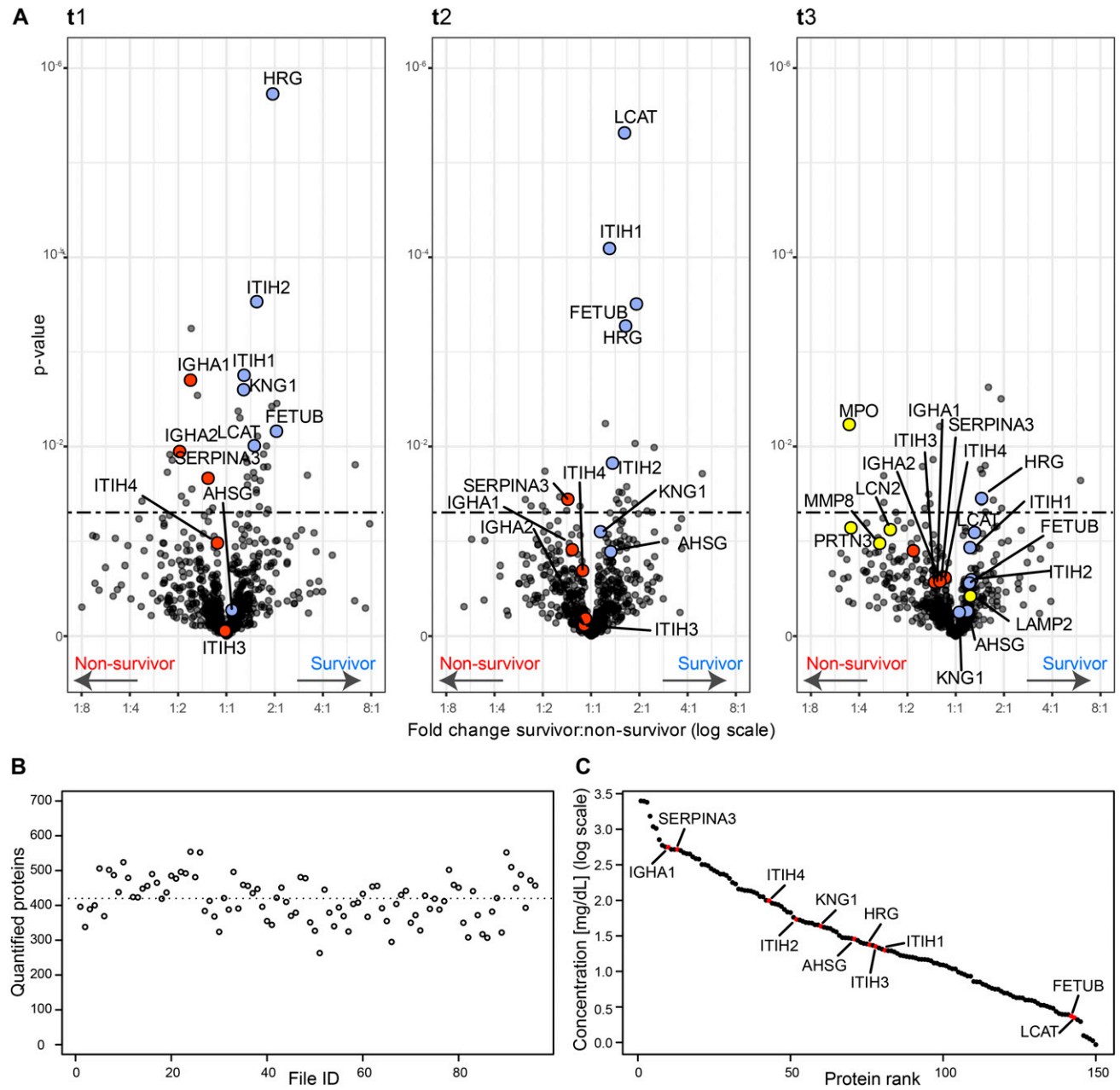

**Figure 2. Serum proteins that are differentially abundant in survivors and non-survivors per time point.**
**(A)** Volcano plots showing the fold change and associated *P*-values. For each time point t1, t2, and t3, a two-sided *t* test was performed to identify the significance of the differentially abundant serum proteins (significance threshold at *P*-value 0.05 indicated as dashed line). Proteins discussed here are represented in red if higher in deceased patients and in blue if showing an increase in surviving patients. In the Volcano plot of t3, some proteins of neutrophil origin are highlighted in yellow. **(B)** The number of quantified proteins per sample, with the dotted line indicating the mean (452 proteins quantified on average). **(C)** The quantitative MS-based data converted to the concentration scale for the 150 abundant serum proteins that are identified and quantified in all samples. Proteins we discuss as potentially stratifying survivors and non-survivors are marked in red (*AHSG, FETUB, KNG1, HRG, ITIH1, ITIH2, LCAT, SERPINA3, IGHA1, IGHA2, ITIH3,* and *ITIH4*) and thus span the entire covered dynamic range.

perform relatively well at time point 2 to stratify between patient outcomes (Fig 3B).

## Discussion

Here, we prospectively performed serum proteomics in moderate-to-severe COVID-19 patients admitted to respiratory and intensive care units because of respiratory failure. Our serum proteomics data provided quantitative information about the abundance of about 300–400 serum proteins on average per patient and per time point, and thus provided quantitative information about changes in protein abundances over disease progression per patient, but also information about serum proteins being more or less abundant when comparing the group of survivors and those that died.

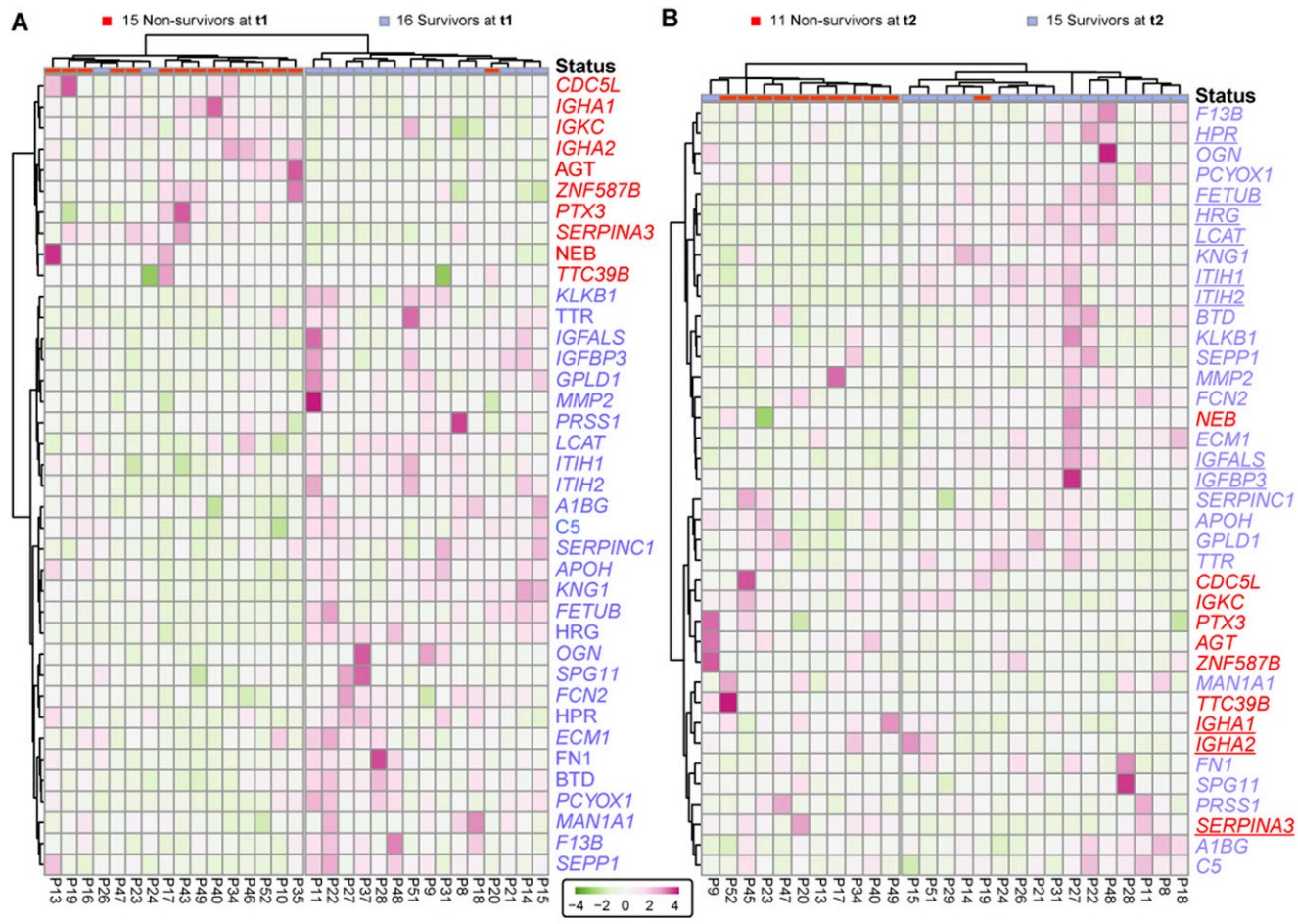

**Figure 3. Surviving and non-surviving SARS-CoV-2 patients can be distinguished by a small panel of abundant serum proteins, already at t1.**
**(A)** Proteins identified as differentially abundant at time point 1 are shown here to completely cluster samples with respect to patient outcome (16 survivors versus 15 non-survivors at time point 1). Proteins are annotated by their gene names, and those indicated in blue show higher concentrations in survivors, whereas proteins in red show higher concentration in non-survivors. **(A, B)** The differentially abundant proteins at time point 1 (as shown in (A)) are used to distinguish the 15 survivors from 11 non-survivors at time point 2. **(B)** Underlined proteins in (B) designate the proteins that were found to be regulated at both time points. Although the cohort consisted of 17 survivors and 16 non-survivors, at each time point, we missed for a few patients a blood sampling point, which then also could not be included in the clustering and is indicated at the top of the dendrograms.

Analyzing the quantitative serum proteomics data, we were intrigued by the fact that several of the highly abundant proteins seemed to be significantly more (or less) abundant in the serum of survivors versus the group of deceased patients. Strikingly, we noted that many of the proteins being more abundant in the survivors were structurally and functionally highly related, such as the type-3 cystatins fetuin-B (*FETUB*), *HRG* and kininogen (*KNG1*), and the the inter-α-trypsin inhibitors (*ITIH1*) and (*ITIH2*). The proteins being more abundant in the deceased patients included α-1-antichymotrypsin (*SERPINA3*), the immunoglobulins IgA (IgA1 and IgA2), the functionally related inter-α-trypsin inhibitors *ITIH3* and *ITIH4*. Moreover, fibronectin (*FN1*) showed a higher abundance in non-survivors, whereas decreasing in survivors. This group of proteins may potentially be considered as candidates for a panel that can be used for mortality risk assessment (Fig 5).

Before determining the optimal composition of such a predictive multi-protein panel, we first discuss in more detail the functional role and relationship of the serum proteins observed to be potentially discriminative between survivors and non-survivors.

*FETUB*, *HRG*, and *KNG1* are in our data, some of the most distinctive differentially abundant serum proteins between survivors and non-survivors, with their levels all about a factor two to four more abundant in the survivors, at almost all three time points sampled. The fourth type-3 cystatin is fetuin-A (*AHSG*), which although not significant in our analysis, followed a similar trend (Fig 4A). Notably, the type-3 cystatins, *FETUB*, *HRG*, *KNG1*, and *AHSG*, are all closely located to each other on human chromosome 3 (Fig 4E).

They also share substantial sequence and domain homology, all containing either two or three alike cystatin domains and *KNG1* and *HRG* share also a His-rich domain. *HRG* is present in human plasma at a concentration of ~75–150 mg/ml in healthy donors and has been implicated in quite a variety of biological functions (20). HRG was also found to be a negative acute-phase reactant, and circulating HRG levels were reported to be significantly lower during acute inflammation and in patients with systemic lupus erythematosus. It has been suggested that HRG may play a critical role in recognizing common molecular "danger" signals in the innate response that protects against tissue damage and pathogen invasion

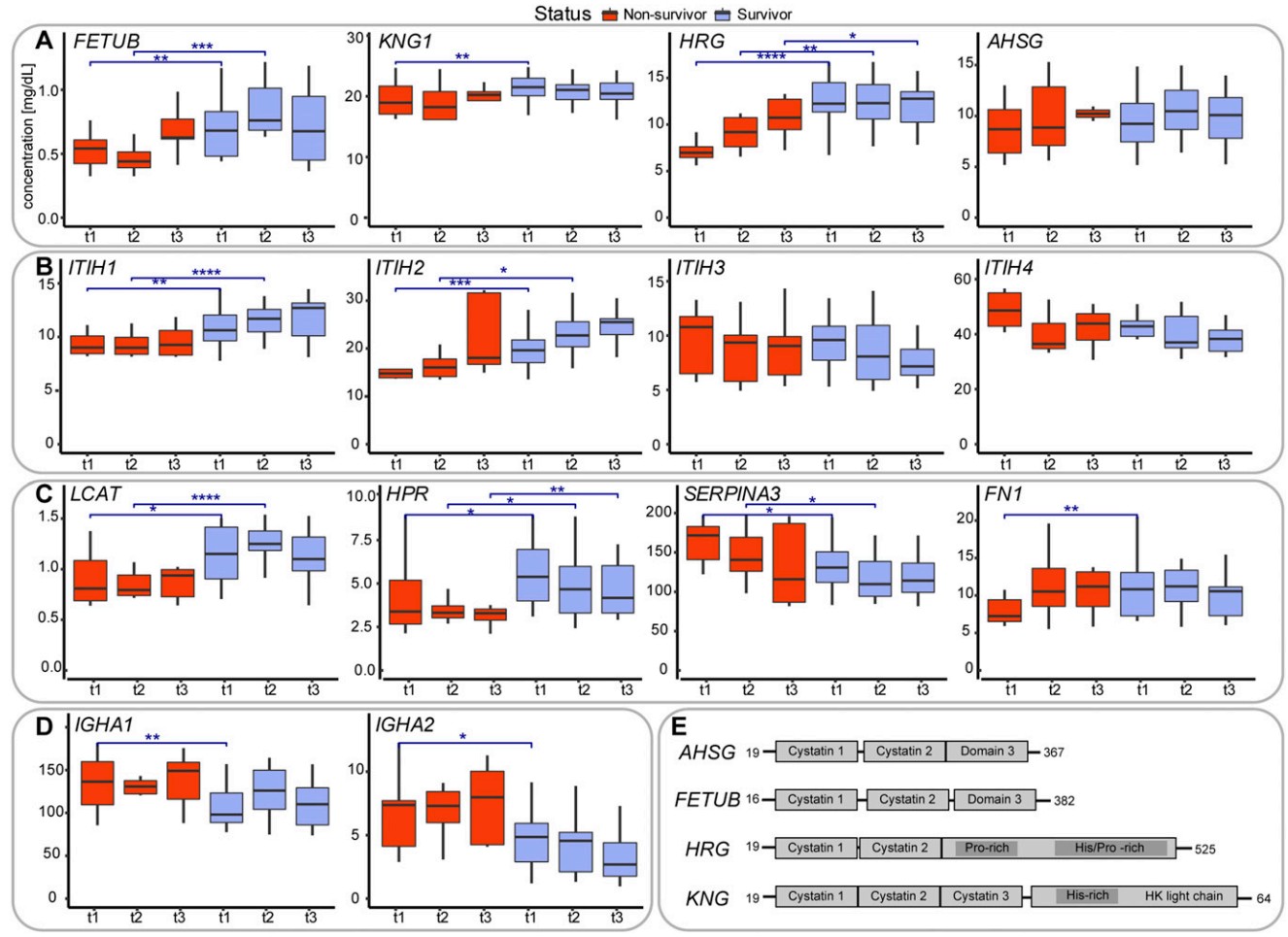

**Figure 4. Serum abundance and structural and functional description of proteins being differentially abundant in survivors versus non-survivors.**
**(A)** Serum abundance estimates, based on MS-based label-free quantification (Fig S2), of the four type-3 cystatins, comparing survivors (blue) with non-survivors (red). At each time point the abundance of these cystatins is higher in the survivors compared to the non-survivors. Comparisons between survivors and non-survivors at respective time points are denoted as significant using asterisks: *$P ≤ 0.05$, **$P ≤ 0.01$, ***$P ≤ 0.001$, and ****$P ≤ 0.0001$. **(B)** Serum abundance of the four abundant inter-α-trypsin inhibitors (IαI), with clear opposing trends between *ITIH1* and *ITIH2* as well as between *ITIH3* and *ITIH4*. At each time point the abundance of *ITIH1* and *ITIH2* is higher in the survivors compared to the non-survivors, whereas for *ITIH3* and *ITIH4* the opposite holds. **(C)** Serum abundance of other putative mortality indicators: phosphatidylcholine-sterol acyltransferase (*LCAT*), HPR, alpha-1-antichymotrypsin (*SERPINA3*), and fibronectin (*FN1*). **(D)** Profiles of the IgA immunoglobulin variants IgA1, IgA2, both less abundant in survivors. **(E)** Schematic domain-structures of the type-3 cystatins showing their sequence homology. Cystatin domains as well as His/Gly and His/Pro domains are depicted as boxes.

as well as aiding wound healing. Still, a clear function of HRG in plasma has not been defined, instead it has been termed an important multifunctional protein, or even Swiss Army knife because of its ability to interact with a wide range of small molecules and other plasma proteins as reviewed by Poon et al. Fetuin-A, referred to as α-2-HS-glycoprotein (*AHSG*), is also an abundant and important plasma protein, albeit also defined as a multifunctional protein (21, 22). *FETUB* is a close paralog of *HRG*; alignment of these two genes reveals 35% identity. Also, its functional role in plasma is still, to some extent, obscure and certainly also multifunctional. Finally, Kininogen-1 (KNG1), also known as α-2-thiol proteinase inhibitor, is best known as the precursor for the low molecular weight peptide bradykinin. It contains three cystatin domains and shares a histidine-rich domain with *HRG*. Thus, *AHSG*, *FETUB*, *HRG*, and *KNG1* share domain structure (cystatin domains), chromosome localization, and functionality. In our serum proteomics data, the abundances of *AHSG*, *FETUB*, *HRG*,

and *KNG1* are found to be highly correlated in each of the sampled COVID-19 patients and are consistently higher in the survivors. In general, it does, however, seem that the abundance levels of these type-3 cystatins remain fairly constant during disease progression.

Markedly, two recent studies have hypothesized that these abundant plasma proteins may indeed represent mortality markers. First, Hong et al (14) used a multiplexed antibody-based affinity proteomics assay to screen 156 individuals aged 50–92 yr. This dataset revealed a consistent age association for HRG. They concluded, by validating this finding in several additional data sets (N = 3,987), that HRG associates with age and risk of all-cause mortality, whereby a genome-wide association study (GWAS) analysis indicated that particular mutations in *HRG* may influence the mortality risk. Second, Wozniak et al (13) analyzed a cohort of around 200 patients by serum proteomics and metabolomics to assess

whether there are potential mortality risk biomarkers for patients who suffered a *S. aureus* bacteremia. The statistically most significant prediction marker they observed was FETUB, which was consistently higher in survivors than in diseased patients. These data and our current data therefore may indicate that high serum levels of the cystatins fetuin-A (*AHSG*), fetuin-B (*FETUB*), *HRG*, and *KNG1* are general positive survival factors, independent of the pathogen that induces the disease.

### The family of inter-α-trypsin inhibitors (IαI)

In our dataset, a group of other functionally related proteins stand out, all belonging to the family of inter-α-trypsin inhibitors. Four of these are abundantly present in our serum dataset, namely, *ITIH1*, *ITIH2, ITIH3*, and *ITIH4*. Notably, *ITIH1* and *ITIH2* are statistically significant differently abundant in survivors and non-survivors, but all four inhibitors also show opposite trends during disease progression (Fig 4B). Notably, the abundance levels of *ITIH1* and *ITIH2* correlate extremely well in each serum sample analyzed, which we actually expected as they are known to form in serum an ~225-kD complex, named IαI, containing Bikunin next to ITIH1 and ITIH2. We should therefore observe a strong positive correlation between *ITIH1* and *ITIH2*, which we do, providing confidence in our quantification. *ITIH1* and *ITIH2* are consistently higher in survivors than in non-survivors. In addition, the levels of these two proteins increase during disease progression, both in the survivors and non-survivors. In sharp contrast, the abundance levels of *ITIH3* and *ITIH4* (that do not form a complex) show a decreasing trend during disease progression, and moreover these proteins are, although just below our statistical cutoff, consistently lower in abundance in survivors than in non-survivors. In a comprehensive recent review by Lord et al (23), the structural organization and functional role of the members of these inter–α-trypsin inhibitors (IαI) is described. The exact role of these proteins in serum is not fully clear, although they all exhibit matrix protective activity through protease inhibitory action. Next, IαI family members interact with the extracellular matrix and most notably hyaluronan, inhibit complement, and provide several cell regulatory functions. Notably, a reduction in circulating IαI has been reported in patients with sepsis which correlated with increased mortality rates (24). Moreover, administration of exogenous IαI has been shown to lead to reduced mortality, suggesting a protective role of specific IαI family members (15). It is somewhat difficult to relate these earlier findings with our data, as in these studies, using a broad-spectrum antibody, no distinction was made between the different inter–α-trypsin inhibitor family members. Still, as in our samples *ITIH4* is by far the most abundant in serum, it is nice to see that its abundance is indeed lower in the non-survivors, consistent with the findings of Lim et al and Opal et al, where inter–α-inhibitor proteins were found to be reduced in severe sepsis, and failure of recovery of IαIp levels over the course of sepsis was associated with mortality. Our data confirm that hypothesis when considering *ITIH4* and *ITIH3*, but notably the opposite is observed for *ITIH2* and *ITIH1*. In summary, the four related inter–α-trypsin inhibitor members possibly provide a panel for monitoring disease outcome in a range of pathogen-caused diseases, among them COVID-19. Of note, in the context of sepsis, IαI improves endothelial inflammation, whereas their levels are inversely associated with

the levels of endothelial dysfunction biomarkers VCAM-1 and ICAM-1 (25). Endothelial dysfunction is a feature of COVID-19 (26), and we have previously observed that high levels of ICAM-1 (27) and VCAM-1 (28) at admission are associated with the mortality of our COVID-19 patients. Our new data suggest that IαI may provide protection from endothelial complications of COVID-19, thereby potentially improving survival.

Finally, IαI has been used in the treatment of inflammatory conditions such as sepsis (one of the most common causes of death for COVID-19 patients [29]) but also for Kawasaki disease, which has recently been associated with SARS-CoV-2 infection in children (30).

### Other putative mortality predictors

Next to the family of type-3 cystatins and the family of inter–α-trypsin inhibitors, only a few more serum proteins did stand out significantly as potential mortality predictors. These included the phosphatidylcholine-sterol acyltransferase (LCAT), an enzyme involved in the extracellular metabolism of plasma lipoproteins, which showed a trend similar to ITIH1 and ITIH2, consistently higher in survivors than in non-survivors and increasing in abundance during disease progression, both in the survivors and non-survivors. Reversely, the family of immunoglobulin A, IgA1 and IgA2, seemed to be more abundant in non-survivors as well as α-1-antichymotrypsin (*SERPINA3*) (Fig 4C and D). Thereby, angiotensin and α-1-antichymotrypsin showed a decrease in abundance over time, whereas the IgA levels seemed to remain more constant over time.

### Predictive power of various multi-protein panels

In our study, several plasma proteins displayed differences in abundance in survivors and non-survivors. Moreover, many of these serum proteins are structurally and functionally related. Therefore, we investigated next whether we could define an optimal multi-protein panel to predict mortality, based on the statistics, but also on the functionality of the proteins and their abundance. Ideally, such a panel is as small as possible but should also have the best predictive power, not only in our cohort and time points but also in independent cohorts.

Given that the *HRG* protein displayed by far the most significant *P*-values in stratifying survivors and non-survivors in our Ferrara cohort, we asked whether *HRG* alone could already predict disease outcome (Fig 5A), or whether a panel of proteins would perform better for predicting mortality. We trained a support vector machine on the data from time point t1 for three different panels: *HRG* alone, a 9-protein panel, and a 16-protein panel (Fig 5A). This model was then used to classify the patients at time points t2 and t3 to assess the model robustness by using receiver operating characteristic (ROC) curves, using 10-fold cross-validation (Fig 5). At time point t1, performance of the one-protein panel (i.e., *HRG*) was quite good with an area under curve (AUC) of 0.92, showing *HRG* alone already has some discriminatory power. *HRG* was found to be less discriminating at time point t2 (AUC 0.85) and time point t3 (AUC 0.78) (Fig 5B). Next, we evaluated the performance of the 9-protein panel, the components of which were selected based on their significance ranking at time point 1 and/or due to their functional relationships

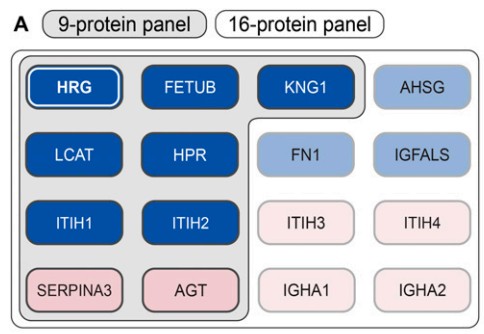

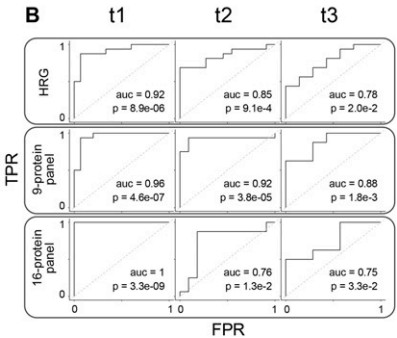

**Figure 5. Performance of different multi-protein panels in predicting mortality.**
**(A)** Definition of the different panels of protein mortality markers (1) histidine-rich glycoprotein (*HRG*) as a single protein biomarker performs best at time point 1. (2) Nine-protein panel consisting of the type-3 cystatins *FETUB*, *KNG1*, and *HRG*, the Inter-α-trypsin inhibitors *ITIH1*, *ITIH2*, and *LCAT*, *HPR*, *SERPINA3*, and *AGT*. (3) Sixteen-protein panel that includes next to the proteins of panel 2 also the type-3 cystatin AHSG, the Inter-α-trypsin inhibitors *ITIH3*, *ITIH4*, the immunoglobulins A1 and A2, fibronectin *FN1*, *IGFALS*, and *AGT*. **(B)** Performance of the different multi-protein panels in our Ferrara cohort at time points 1, 2, and 3, depicted by using ROC curve analysis, revealing that the multi-protein panels outperform the single protein *HRG* as a mortality predictor. The 9-protein panel performs best, especially at the earliest time points just shortly after critically ill patients have entered the ICU.

(Fig 5A). This panel displayed improved predictive power in comparison to the one-protein HRG panel, providing in the ROC analysis AUCs of 0.96, 0.92, and 0.88 at time points t1 (training set), t2, and t3, respectively. Finally, we evaluated a third panel consisting of 16 proteins (Fig 5A), extended by the inclusion of more functionally related serum proteins. This panel performed even better at time point 1, with an AUC of 1, but this panel did not keep its predicting power at the later two time points t2 and t3, whereby the AUCs diminished to 0.77 and 0.71, respectively.

## Comparison with related recent COVID-19 plasma proteome studies

Although the panel of plasma protein markers we here define as putative mortality indicators is very significant in stratifying the survivors from the non-survivors affected by COVID-19, our study has a few limitations. First of all, it relates to a rather small cohort of patients. Moreover, patients' characteristics mainly related to medical history and treatments can affect the measured outcomes. However, we also believe that the prospective longitudinal design adopted in the study and the rigorous clinical follow-up increase the strength of the results, particularly in relation to clinical outcomes. With this in mind, our study should ideally be compared to alike plasma proteomics studies on other independent cohorts. In the last months, the research efforts on COVID-19 have expanded enormously. In this period, quite a few multi-omics and plasma proteomics studies have appeared studying COVID-19 patients, generating data comparable to ours, but with different research questions and thus also different study designs (2, 5, 6, 7, 8 *Preprint*, 31 *Preprint*). Still, the outcome of these studies and their conclusions can be compared with the data obtained in our cohort (further termed "Ferrara cohort"). We made a selection, focusing on recent studies most relevant to our data and compare their findings with ours below.

First of all, Demichev et al in a study coordinated by the PA-COVID-19 study group (8 *Preprint*) measured the plasma proteome of 139 hospitalized patients at the Charité–Universitätsmedizin Berlin and followed the time-dependent progression of COVID-19 through different stages of the disease, combining several diagnostic clinical parameters and plasma proteomes (here further termed "Berlin cohort"). They used clinical parameters to classify the patients in cohorts of increasing severity and followed changes in the

patients' clinical parameters and plasma proteomes over time. From their data, they were able to define signatures for disease states as well as observe age-related plasma proteome changes. Relevant to our work, they report that low plasma levels of 54 proteins could be associated with disease severity. Comparing that list with the list of 28 proteins that are found to be significantly lower in non-survivors versus survivors at t1 in the Ferrara cohort, we observe a large overlap, including *HRG*, *FETUB*, *AHSG*, *IGFALS*, *GPLD1*, *LCAT*, *TTR*, *SERPINC1*, *HPR*, *ITIH1*, and *ITIH2*, all lower in abundance in the plasma of severe COVID-19 patients. The list of 54 proteins of Demichev et al is larger than ours, but their dataset also includes proteins we disregarded as RBC contaminants, such as variants of hemoglobin (*HBD*, *HBB*, and *HBA1*) and carbonic anhydrase, or excluded from our analysis for other reasons, such as albumin. Reversely, we also examined the list of proteins they state as being of high abundance in the plasma of patients with poor prognosis in the Berlin cohort and also observe a substantial overlap. As was the case for low abundant proteins in patients with poor prognosis, their list includes proteins we disregarded, such as the fibrinogens *FGA*, *FGB*, and *FGG*, as these levels may be affected by the sampling. Other proteins they note are also in our list of proteins being higher in non-survivors at t1, notably *SERPINA3*, *AGT*, *ITIH3*, and *ITIH4*.

From the Berlin clinical and plasma proteome data, the authors ultimately defined a very narrow panel of proteins predicting future worsening of COVID-19 disease. The four proteins defined by them to be indicative of poor prognosis when their plasma levels are low were *AHSG*, *HRG*, *ITIH2*, and *PLG*. Pleasingly, except for *PLG*, all these proteins were also found by us to be substantially lower in the Ferrara non-survivors than non-survivors. Surprisingly, *ITIH1* was not mentioned by Demichev et al in this panel, although the protein levels of this protein are known to directly correlate with that of *ITIH2*. Conversely, they defined seven proteins whereby high abundance in plasma would be indicative of poor prognosis, including *SERPINA3* and *AGT*. Both these latter proteins are also part of the small putative mortality panel we define here and are indeed lower in abundance in Ferrara non-survivors. The data obtained for the Berlin cohort are therefore in excellent agreement with our findings. To illustrate this agreement more quantitatively, we show that our 9- and 16-protein panels also show clear differences in patient survival in the Berlin cohort (Fig S5A and B). To maintain a fair comparison of survivors and non-survivors, we needed to select a subset of relevant patients from the Berlin cohort. First, we

selected those patients falling under World Health Organization (WHO) category 6 or higher, which denoted receipt of invasive mechanical ventilation, indicating an acute phase. This was done to match with the state of patients in our Ferrara cohort, who were all admitted to the intensive care unit (ICU). For each of these patients, we selected the first and last time points available within the WHO category (in some cases, the last time point corresponded to the day of death). Furthermore, filtering was carried out to remove patients with do-not-intubate/do-not-resuscitate (DNI/DNR) orders as well as those not yet discharged from hospital as far as was known to the authors at the end point of their study. Focusing on the panel of 16 proteins defined in Fig 5, it may be observed that in the Berlin cohort, almost all these 16 proteins display similar trends as observed in the Ferrara cohort, despite the patients and study design being very different (Fig S5B). We next used our three defined panels (HRG, the 9-protein panel, and the 16-protein panel) to probe the extent to which these can discriminate between survivors and non-survivors in the Berlin cohort (Fig S6A) and observed that both the 9- and 16-protein panel provide very good stratification with high AUCs of all >0.85.

Another recent plasma proteome profiling study related to ours has been reported by Geyer et al (31 Preprint). They primarily studied differences between the plasma proteomes of 31 COVID-19 patients versus 263 PCR-negative controls, from the University Hospital of LMU Munich (here termed "Munich cohort"). In that analysis, Geyer et al found that the protease inhibitors SERPINA3, ITIH3, and ITIH4 were increased in plasma abundance in COVID-19 patients, when compared with controls, whereas the HRG and fibronectin (FN1) were decreased. In the Ferrara study, we did not compare the COVID-19 plasma proteomes to PCR-negative controls but focused on ICU-hospitalized survivors versus non-survivors. Still, we also find that in the survivors, SERPINA3, ITIH3, and ITIH4 were lower in abundance and HRG and FN1 higher in abundance, when compared with the non-survivors, confirming that low levels of HRG are potentially a hallmark of COVID-19 disease diagnosis and disease severity.

Next, Geyer et al also followed the protein abundances in the 31 COVID-19 patients over time in their Munich cohort. They measured longitudinal trajectories of 116 proteins (a list including many immunoglobulin variants, which we chose to exclude in our analysis) that significantly changed over a disease course of up to 37 d. Their study, albeit with a similar size cohort, but with more frequent sampling at a greater number of time points, cannot be directly compared with our data for the 33 patients from Ferrara, sampled at just three time points. However, it is striking and pleasing to see that the trends in protein abundance observed are in very good agreement, certainly for the small panel we define here as putative mortality markers. Pleasingly the trends observed for all the four abundant inter-α-trypsin inhibitors (ITIH1, ITIH2, ITIH3, and ITIH4) are alike in both studies, with an increase over time for the first two and a decline in the latter two (Fig S5A and C). Moreover, the plasma proteins HRG, FETUB, KNG1, LCAT, AHSG, and FN1 increase over time in abundance in the Munich cohort, thus suggesting their regulation during COVID-19 disease development. Finally, SERPINA3, found to decrease over time in our study was observed to decrease over time in the Munich cohort as well. We also aimed to use the Munich cohort data from Geyer et al to test the validity of our protein panels. To do so, we selected a subset

matching in age and disease status with our cohort from survivors from non-survivors at discrete time points. We thus selected only persons tested positive for COVID-19 (thus disregarding healthy control subjects) and further selected only three time points, 1, 7, and 14 d after hospital admission. With this step, we aimed to match this data as much as possible to the time points t1, t2, and t3 of the Ferrara cohort. We show that also in this second independent cohort analyzed by Geyer et al, the trends of our 9- and 16-protein panel are in good agreement with the data obtained for the Ferrara cohort (Fig S5C). Moreover, we used again our three defined panels (HRG, the 9-protein panel, and the 16-protein panel) to probe the extent to which these can discriminate between survivors and non-survivors in the Munich cohort (Fig S6B) and observed that both the 9- and 16-protein panel provide very good stratification at the time point 7 d after admission with high AUCs, all of >0.85, but with weaker performance at the other time points. It should be noted that the number of non-survivors included in our analysis of the Geyer data is only a handful (n = 5–6), which may be a cause for the somewhat less strong correlation found.

In addition, relevant to our work, Galbraith et al (6) reported a very extensive multi-omics study, performing in parallel plasma proteomics by mass spectrometry and SOMAscan assays, cytokine profiling, and immune cell profiling via mass cytometry. They compared 32 controls with 73 COVID-19 positive donors. They classified the plasma of the donors infected by COVID-19 by their seroconversion status and made the primary conclusion that the timespan between exposure to the virus and seroconversion could be a key determinant of disease severity. Classifying seroconversion in a low and a high category, they found several molecular determinants to be correlated. Most related to our work, in the high seroconversion category, several proteins were found to be differentially abundant. Among others, these included several complement related proteins (which are not statistically differential in our data). Other proteins observed to be significantly differential in the plasma proteomics and SOMAscan analyses of Galbraith et al (6) include SERPINA3, SERPINC1, PLG, and KNG1, which follow the same trend as in the data of Geyer et al (31 Preprint), Demichev et al (8 Preprint) as well as our work. Overall, our data is in better agreement with the Geyer et al and Demichev et al data sets, but that may also be due to the fact that the classification made by Galbraith et al was based on seroconversion, and thus different from the classifications made by the other groups.

Therefore, although the study designs as well as the patient cohorts were completely different, our protein panels predicting mortality perform extremely well in other cohorts. The huge consistency in the findings provides credibility to these independently made observations, especially considering the still modest number of patients all three studies tackled.

In summary, in our study comparing one by one the serum proteomes of a closely matched group of survivor and non-survivor COVID-19 patients admitted to the ICU, we were able to extract a small panel of proteins that can be used to predict the disease outcome, already at an early stage, that is, shortly after admission to the ICU. The ability to stratify patients at a single time point so soon after hospital admission provides in our view essential information highly relevant to clinicians. This set of mortality indicators consists largely of functionally related proteins, namely,

several type-3 cystatins and the family of inter-α-trypsin inhibitors. Although our patient cohort was rather small, our observations and conclusions are in pleasing agreement with two other, independent, recent serum proteome studies, which basically define the same set of proteins as either markers or disease severity or mortality.

# Materials and Methods

### Ferrara patient cohort serum sample collection and chemicals

The present analysis included patients from the "Pro-thrombotic status in patients with SARS-CoV-2 infection" (ATTAC-Co) study (32, 33). The ATTAC-Co study is an investigator-initiated, prospective, single-centre study recruiting consecutive patients admitted to hospital (University Hospital of Ferrara, Italy) because of respiratory failure due to COVID-19 between April and May 2020. Inclusion criteria were (i) age >18 yr; (ii) confirmed SARS-CoV-2 infection; (iii) hospitalization for respiratory failure; (iv) need for invasive or non-invasive mechanical ventilation or only oxygen support. Exclusion criteria were prior administration of P2Y12 inhibitor (clopidogrel, ticlopidine, prasugrel, and ticagrelor) or anticoagulant drugs (warfarin or novel oral anticoagulants), known disorder of coagulation or platelet function and/or chronic inflammatory disease. SARS-CoV-2 infection was confirmed by RT-PCR assay (Liaison MDX; Diasorin) from nasopharyngeal swab specimen. Respiratory failure was defined as a PaO2/FiO2 (P/F) ratio ≤200 mmHg. Clinical management was in accordance with current guidelines and specific recommendations for COVID-19 pandemic by Health Authorities and Scientific Societies. Three different blood samplings were made; just after admission (t1), after 7 ± 2 d (t2), and after 14 ± 2 d (t3). Study blood samplings were performed from an antecubital vein using a 21-gauge needle or from central venous line. All patients underwent blood sampling in the early morning, at least 12 h after last administration of anticoagulant drugs. The first two to 4 ml of blood was discarded, and the remaining blood was collected in tubes for serum/plasma collection. The serum and plasma samples were stored at –80°C. The planned blood sample withdrawals were not performed in case of patient's death or hospital discharge. The ATTAC-Co study population consists of 54 moderate-to-severe COVID-19 patients (32, 33). The subgroup of interest for the present analysis is selected starting from the 16 cases who died. From the remaining 38 survivors, we identified 17 cases who best matched in terms of age, clinical history, and clinical presentation. This selection was done with the aim to maximize the possibility to identify differences between deceased and survivors and minimizing potential confounding factors. The protocol was approved by the corresponding Ethics Committee (Comitato Etico di Area Vasta Emilia Centro, Bologna, Italy). All patients gave their written informed consent. In case of unconsciousness, the informed consent was signed by the next of kin or legal authorized representative. The study is registered at www.clinicaltrials.gov with the identifier NCT04343053.

### Serum sample preparation

24 μl of a detergent-based buffer (1% sodium deoxycholate (SDC), 10 mM tris(2-carboxyethyl)phosphine (TCEP), 10 mM Tris, and 40 mM chloroacetamide) with Complete mini EDTA-free protease inhibitor cocktail (Roche) was added to 1 μl serum to enhance protein denaturation and boiled for 5 min at 95°C. 50 mM ammonium bicarbonate was added and digestion was allowed to proceed overnight at 37°C using trypsin (Promega) and LysC (Wako) at 1:50 and 1:75 enzyme:substrate ratios, respectively. The digestion was quenched with 10% formic acid and the resulting peptides were cleaned-up in an automated fashion using the AssayMap Bravo platform (Agilent Technologies) with corresponding AssayMap C18 reverse-phase column. The eluate was dried and resolubilized in 1% FA to achieve a concentration of 1 μg/μl, of which 1 μL was injected.

### Serum proteome profiling

All spectra were acquired on an Orbitrap HFX mass spectrometer (Thermo Fisher Scientific) operated in the data-independent mode (DIA) coupled to an Ultimate3000 liquid chromatography system (Thermo Fisher Scientific) and separated on a 50-cm reversed phase column packed in-house (Poroshell EC-C18, 2.7 μm, 50 cm × 75 μm; Agilent Technologies). Proteome samples were eluted over a linear gradient of a dual-buffer setup with buffer A (0.1% FA) and buffer B (80%ACN, 0.1%FA) ranging from 9 to 40% B over 95 min, 40–55% B for 4 min, 55–95% B in 1 min, and maintained at 95% B for the final 5 min with a flow rate of 300 nl/min. DIA runs consisted of a MS1 scan at 60,000 resolution at m/z 200 followed by 30 sequential quadrupole isolation windows of 20 m/z for HCD MS/MS with detection of fragment ions in the orbitrap (OT) at 30,000 resolution at m/z 200. The m/z range covered was 400–1,000 and the Automatic Gain Control was set to $1 \times 10^6$ for MS and $2 \times 10^5$ for MS/MS. The injection time was set to 100 ms for MS and "auto" for MS/MS scans.

### Data analysis

Spectra were extracted from the DIA data using DIA-NN (version 1.7.12) without a spectral library and with "Deep learning" option enabled. The enzyme for digestion was set to trypsin and one missed cleavage was tolerated. Cysteine Carbamidomethylation and Methionine oxidation were allowed. The precursor false discovery rate threshold was set to 1%. Protein grouping was done by protein names and cross-run normalization was RT-dependent. All other settings were kept at the default values. The gene-centric report from DIA-NN was used for downstream analysis, and quantification was based on unique peptides. When injection replicates were available, the median of these values was used. All downstream analyses were carried out in R (34). For all proteins, the concentration was estimated based on a set of reference proteins using a log–log model (Fig S2). Significant proteins were determined using results from a two-sided t test, with a P-value cutoff of $5 \times 10^{-2}$. The protein panels were chosen based on significant values at time point 1, complemented by a few protein proteins being structurally and functionally related. To estimate the ability to categorize patient outcome, we constructed ROC curves based on a linear support vector machine (SVM) model fitted onto the first time point of the data, fitted onto the later time points. Reliability was determined with 10× cross-validation where possible. SVMs were fitted

using the e1071 R package (35), and results were plotted using the pROC package (36). AUC and *P*-value were estimated using the Wilcoxon–Mann–Whitney U test. Unsupervised hierarchical clustering was performed using Ward's algorithm with euclidean clustering distance.

## Data Availability

The mass spectrometric proteomics data have been deposited at the ProteomeXchange Consortium via the PRIDE partner repository (37) with the dataset identifier PXD024707.

## Supplementary Information

## Acknowledgements

We acknowledge support from the Dutch Research Council (NWO) funding the Netherlands Proteomics Centre through the X-omics Road Map program (project 184.034.019) and the EU Horizon 2020 program INFRAIA project Epic-XS (Project 823839).

### Author Contributions

F Völlmy: data curation, formal analysis, investigation, methodology, and writing—original draft, review, and editing.
H van den Toorn: data curation, software, formal analysis, investigation, methodology, and writing—review and editing.
R Zenezini Chiozzi: investigation and project administration.
O Zucchetti: resources, investigation, project administration, and writing—review and editing.
A Papi: resources and formal analysis.
CA Volta: resources and formal analysis.
L Marracino: resources and formal analysis.
F Vieceli Dalla Sega: resources, data curation, formal analysis, and investigation.
F Fortini: resources and formal analysis.
V Demichev: resources, formal analysis, and writing—review and editing.
P Tober-Lau: data curation and formal analysis.
G Campo: resources, data curation and formal analysis.
M Contoli: conceptualization, resources, formal analysis, supervision, and writing—review and editing.
M Ralser: resources, formal analysis, supervision, investigation, and writing—review and editing.
F Kurth: resources, supervision, and writing—review and editing.
S Spadaro: data curation, formal analysis, investigation, and writing—review and editing.
P Rizzo: conceptualization, supervision, and writing—review and editing.

AJR Heck: conceptualization, data curation, formal analysis, supervision, funding acquisition, investigation, and writing—original draft, review, and editing.

### Conflict of Interest Statement

The authors declare that they have no conflict of interest.

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
