## [Reviewer comments · Life Science Alliance]

Life Science Alliance

A serum proteome signature to predict mortality in severe COVID-19 patients

Franziska Voellmy, Henk van den Toorn, Riccardo Zenezini Chiozzi, Ottavio Zucchetti, Alberto Papi, Carlo Volta, Luisa Marracino, Francesco Vieceli Dalla Sega, Francesca Fortini, Vadim Demichev, Pinkus Tober-Lau, Gianluca Campo, Marco Contoli, Markus Ralser, Florian Kurth, Savino Spadaro, Paola Rizzo, and Albert Heck

DOI: <https://doi.org/10.26508/lsa.202101099>

Corresponding author(s): Albert Heck, Utrecht University

Review Timeline:

Submission Date:	2021-04-16
Editorial Decision:	2021-05-19
Revision Received:	2021-06-11
Editorial Decision:	2021-06-22
Revision Received:	2021-06-24
Accepted:	2021-06-24

Transaction Report:

Please note that the manuscript was previously reviewed at another journal and the reports were taken into account in the decision-making process at Life Science Alliance.

Referee #1 Review

Comments on Novelty/Model System for Author:

The title and the conclusion make/imply a claim that a serum proteome signature could "predict mortality in severe COVID-19 patients", however, the key results as shown in Fig3 display only preliminary data of unsupervised clustering of 17 survivors and 16 non-survivors sera samples based on expression of a list of proteins. There is no model composed of a "panel" to predict the mortality and no measure of the prediction performance, and no independent validation of the panel. Instead, the data in essence show that some sera proteins are different between 17 survivors and 16 non-survivors, and some of these differentially expressed proteins are also reported in the literature when they compared severe versus non-severe cases.

Remarks for Author:

The author discovered some serum protein markers differentially expressed, at 2-3 time points, between 17 survivors and 16 non-survivors during the pandemic. They make/imply a claim that eight proteins, complemented with a few more, may represent a panel for mortality risk assessment and eventually even for treatment. While they studied very previous COVID-19 specimens, the authors may consider further improve the manuscript in the following aspects.

Major issues:

- 1) What is exactly the panel for predicting COVID-19 mortality? What are the proteins in the panel and how the panel makes a prediction? What is the performance of this panel in terms of sensitivity, specificity, accuracy and ROC, in the patients studied here? How about the performance in independent patient cohorts/data sets?
- 2) To identify the differentially expressed proteins, have the student t tests been corrected taking into account multiple testing? Have they set any threshold for fold-change?
- 3) There are multiple serum proteomics datasets from COVID-19 patients in the literature, the authors mentioned some of them in the Discussion. A comprehensive, quantitative and comparative analysis of their data to the data sets from the literature should be performed to evaluate the value of the "panel", considering the sample size of this manuscript is too small.
- 4) Figure 4: the concentrations of these proteins were estimated based on the literature with certain degree of accuracy (the overall $R^2=0.78$), instead of direct measurement. It is misleading to directly use the absolute quant "mg/dL" as the y-axis.

Minor issues:

- 1) It is necessary to perform a more comprehensive literature review of the blood molecular (including protein) profiling of the COVID-19 patients in the Introduction.
- 2) Typo: in some cases "COVID-19" was incorrectly spelt as "COVID19". So does "SARS-CoV-2".
- 3) Have the sera been de-activated to avoid potential infection? How much microliter sera was processed for proteomic analysis? How much peptides were injected into the MS?

Referee #3 Review

Remarks for Author:

In this manuscript the authors have measured serum protein levels by mass spectrometry for samples collected from 33 individuals infected with SARS-CoV-2 (17 survivors, 16 deceased) at one to three timepoints following admission. On the order of ~450 proteins were quantified and several were found to be significantly different between patients that survived or not. The biomarkers of disease severity or mortality were then compared with biomarkers for COVID-19 severity identified in other studies (Geyer et al. 2021 and Demichev et al. 2020). To the authors own admission, this is a rather small cohort and there are important issues that make the evaluation of this result challenging even if one considers this work on its merit as a replication study. As it stands I find it difficult be able to draw strong conclusions from the manuscript and also unsure about its value within the context of other related advances.

Major concerns

1 - I have concerns about the statistical analysis that were performed and the extent the authors can make the claims they do with the current dataset. There are two main issues, one has to do with the characterization of the cohort and potential confounders and the other more simply to do

with multiple hypothesis testing.

1.1. - The authors tried to match to their best capacity the cohorts which is clear in regards to age and gender. There seems to be some difference in age that could already be important if they were to consider it in their statistical modelling. There may still be other aspects such as other underlying conditions that are not fully explained in the manuscript that may also need to be considered. In this regard, when comparing the two groups of patients and testing for the significant predictive value of a specific protein in distinguishing between them it is usually needed to consider such other factors (age, etc) as confounder. This would need to be achieved using a more sophisticated statistical test.

1.2 - In addition to description and incorporation of confounder into their statistical testing, the authors also need to account for multiple hypothesis testing (i.e. the number of tested proteins). The p-value cut-off used in the manuscript does not appear to take this into account. A stringent cut-off that considers 450 tests would put the p-value at 10^{-4} . Looking at the plots it would seem that there would be very few things that would be deemed significant. Considering the potential impact of accounting for the co-founders this may lead to no result being deemed significant.

2 - This work is focused on deriving protein biomarkers for COVID-19 mortality or severity. This is most related to two preprints (Geyer et al. 2021 and Demichev et al. 2020) and a recently published study (Galbraith et al. eLife 2021). While not strictly focused on comparing survivors from deceased these other works are larger in patient numbers and provide much of the value in terms of biomarker discovery and replication. While I certainly believe the topic to be of high importance and I am unsure about what this manuscript and work would add beyond what those already cover. Given the statistical issues from point 1, I have a difficult time in suggesting an application of the current data and manuscript that stands on its own and provides a significant advance in this area. If a predictive model was built on other larger datasets, then this cohort would perhaps serve as a useful independent test set but on its own it seems currently limited.

May 19, 2021

Re: Life Science Alliance manuscript #LSA-2021-01099-T

Prof. Albert J.R. Heck
Utrecht University
Biomolecular Mass Spectrometry and Proteomics Group
Padualaan 8
Utrecht 3584 CH
Netherlands

Dear Dr. Heck,

Thank you for transferring your manuscript entitled "Is there a serum proteome signature to predict mortality in severe COVID-19 patients?" to Life Science Alliance.

For a brief overview, the manuscript was previously reviewed at a partner journal, and the authors have chosen to transfer the study along with the reviewer reports to Life Science Alliance (LSA). We think an appropriately revised manuscript in accordance to the points below, could be considered further at LSA.

- + the issues pertaining to statistical analyses, raised by both Rev 1 and 3 (Rev 1 pt 2, Rev 3 pt 1)
- + a model composed of a panel to predict COVID19 mortality is included and further analysis are done to determine the performance of this panel in terms of sensitivity, specificity, accuracy and ROC, in the patients studied in the current manuscript (Rev 1 pt 1)
- + a comprehensive, quantitative and comparative analysis is performed of the data in the current manuscript and the available COVID19 patients serum proteomics data sets from the literature is included (Rev 1 pt 3)
- + Rev 1 pt 4 and minor concerns are addressed
- + please include a point-by-point rebuttal to the reviewers' comments

We, thus, encourage you to send us a revised version of the manuscript that addresses the above mentioned points. The revised version may need to be re-reviewed, in which case we will try to contact the same set of reviewers and walk them through this transfer process.

The typical timeframe for revisions is three months. Please note that papers are generally

considered through only one revision cycle, so strong support from the referees on the revised version is needed for acceptance.

Thank you for this interesting contribution to Life Science Alliance. We are looking forward to receiving your revised manuscript.

Sincerely,

Shachi Bhatt, Ph.D.
Executive Editor
Life Science Alliance
<http://www.lsajournal.org>
Tweet @SciBhatt @LSAJournal

- A letter addressing the reviewers' comments point by point.
- An editable version of the final text (.DOC or .DOCX) is needed for copyediting (no PDFs).
- High-resolution figure, supplementary figure and video files uploaded as individual files: See our detailed guidelines for preparing your production-ready images, <https://www.life-science-alliance.org/authors>
- Summary blurb (enter in submission system): A short text summarizing in a single sentence the study (max. 200 characters including spaces). This text is used in conjunction with the titles of papers, hence should be informative and complementary to the title and running title. It should describe the context and significance of the findings for a general readership; it should be written in the present tense and refer to the work in the third person. Author names should not be mentioned.

B. MANUSCRIPT ORGANIZATION AND FORMATTING:

*****IMPORTANT:** It is Life Science Alliance policy that if requested, original data images must be made available. Failure to provide original images upon request will result in unavoidable delays in publication. Please ensure that you have access to all original microscopy and blot data images

before submitting your revision.**

Comments from (in black) and responses to (in blue) the editor and reviewers

From the editor:

The issues pertaining to statistical analyses, raised by both Rev 1 and 3 (Rev 1 pt 2, Rev 3 pt 1)

We are aware that the multiple-testing corrected p-values (q-values) reveal just a few proteins as differentially expressed in the data. We'd like to stress that we were looking for an optimal panel of protein expression values that can be used to predict mortality. We selected this panel based on uncorrected p-values, and extended the list based on protein homology, structure/function and trends in expression. To address the reviewers concerns further, we have performed additional analyses and revised our manuscript, using in our analysis a machine learning algorithm, a Support Vector Machine, and calculating the FPR/TPR relation showing receiver operating characteristic (ROC) curves. Especially the proposed 9-protein panel results in a consistently high AUC. We have added these panels and the extensive analysis in Figure 5 of the revision.

- A model composed of a panel to predict COVID19 mortality is included and further analysis are done to determine the performance of this panel in terms of sensitivity, specificity, accuracy and ROC, in the patients studied in the current manuscript (Rev 1 pt 1)

In response, we now have performed additional analyses and defined a one-protein panel (HRG), a 9-protein panel and a 16-protein panel and assessed how these panels performed in our own patient cohort at the three time windows that serum was taken. These panels are both based on the statistics of our data as well as on the functional relationship between the proteins within the panel. For instance, in the 16-protein panel, we included all four plasma cystatins and all four plasma inter-alpha-trypsin inhibitors. Our analysis shows that all these panels perform well within our Ferrara cohort, with the best performing panel being the 9-protein panel, which also performs best at the earliest time after hospital admission. As it is important to assess the treatment plan as soon as possible this is also clinically the preferred choice. We included in the revision the definition of these panels (Figure 5) and show the corresponding ROC curves in the new figure 5 and provide the p-values for all proteins taking into these panels in the new supplementary figure 5 and ROC curves in Supp Figure 6.

- A comprehensive, quantitative and comparative analysis is performed of the data in the current manuscript and the available COVID19 patients serum proteomics data sets from the literature is included (Rev 1 pt 3)

We have taken this comment very seriously and requested the data from the Demichev and Geyer studies. We would like to thank these groups for making the data from their MedRxiv studies available to us. We have assessed the serum abundance of all proteins defined in our nine protein and sixteen protein panel in the Munich (Geyer) and Berlin (Demichev) cohorts, for which we have taken time points that were close to our three time windows of serum collection, i.e. timepoint 1 shortly after submission to the ICU, time point 2 on average 7 days after admission and timepoint 3 with an average of 14 days after admission. We also filtered that data taking only severe state patients (WHO > 6) to match with the state of patients in our Ferrara cohort, all admitted to the ICU. Although that meant that these cohorts were reduced to a similar size as our Ferrara cohort (or even smaller), we did observe that nearly all proteins from our defined 9-protein and 16-protein panel, at nearly all these timepoints, showed similar trends as in our study. This we found extremely rewarding.

In the Demichev study most of the proteins also showed significant p-values. In the Geyer study the trend was agreeable, but the statistics weaker, primarily due to a lower number of patients that could be included following the outlined criteria. The data of all these analyses with protein abundance and p-values given for all proteins in the 16-protein panel are now included in the Supplementary figure 5. Based on their support and input we have now included four new co-authors from the laboratories of Ralser and Kurth.

Of note, our approach and analysis strategy are quite different from the published analyses performed by Geyer and Demichev. While their analysis focused primarily on trends in protein abundance longitudinally, while patients were hospitalized, our analysis focused on differences in protein abundance between survivors and non-survivors at a single point in time. Simply put, in the Geyer and Demichev work, a protein has a very strong p-value if the longitudinal trend in abundance is statistically strong, while in our work a protein has a strong p-value when the protein abundance between survivors and non-survivors is substantial and reproducible in all patients. Reanalyzing the Geyer and Demichev data with our approach leads to smaller single timepoint p-values for the proteins compared to the significance values reported in the original studies, which still have very low (i.e. good) p-values when the trend in the concentration over time is taken.

Still the huge reproducibility in the trends seen for especially our nine- and sixteen-protein panel, as shown in the new supplementary figure 5 and 6, really indicates the strength of our panel, and moreover shows that results and conclusions from our plasma proteome data are highly reproducible, even when the data is generated from other cohorts and by other researchers in different countries.

- Rev 1 pt 4 and minor concerns are addressed

We addressed all minor concerns.

Further validation of these findings in an independent patient cohort will not be required for consideration of this study in LSA.

As we now did reach out to the researchers from related studies and did quite a bit more than initially requested by the editor, by first defining mortality predicting panels, evaluating how these panels performed on our own cohort, but also evaluating how the performance of these panels could be reproduced in not just one other cohort, but even two independent cohorts, from different countries and different researchers we feel that our results have even become more important. Clinicians who must make decisions soon after patient admission to the hospital, on who to treat and how to treat them, would ideally assess a very small set of proteins and ideally at a very early single time point. The here defined nine-protein panel will potentially offer just that, and therefore we sincerely hope and would appreciate that the work will be reconsidered for Embo Mol Med.

***** Reviewer's comments *****

Referee #1 (Comments on Novelty/Model System for Author):

The title and the conclusion make/imply a claim that a serum proteome signature could "predict mortality in severe COVID-19 patients", however, the key results as shown in Fig3 display only preliminary data of unsupervised clustering of 17 survivors and 16 non-survivors sera samples based on expression of a list of proteins. There is no model composed of a "panel" to predict the mortality and no measure of the prediction performance, and no independent validation of the panel. Instead,

the data in essence show that some sera proteins are different between 17 survivors and 16 non-survivors, and some of these differentially expressed proteins are also reported in the literature when they compared severe versus non-severe cases.

We now have performed additional analyses and defined a one-protein panel (HRG), a 9-protein panel and a 16-protein panel and assessed how these panels performed in our own patient cohort at the three time windows that serum was taken. These panels are both based on the statistics of our data as well as on the functional relationship in between the proteins within the panel. For instance, in the 16-protein panel, we included all four plasma cystatins and all four plasma inter-alpha-trypsin inhibitors. Our analysis shows that all these panels perform well within our cohort, with the best performing panel being the 9-protein panel, which also performs best at the earliest time after hospital admission. As it is important to assess the treatment plan as soon as possible this is also clinically the preferred choice. We included in the revision the definition of these panels (Figure 5) and show the corresponding ROC curves in the new figure 5 and provide the p-values for all proteins taking into these panels in the new supplementary figure 5.

Referee #1 (Remarks for Author):

The author discovered some serum protein markers differentially expressed, at 2-3 time points, between 17 survivors and 16 non-survivors during the pandemic. They make/imply a claim that eight proteins, complemented with a few more, may represent a panel for mortality risk assessment and eventually even for treatment. While they studied very previous COVID-19 specimens, the authors may consider further improve the manuscript in the following aspects.

Major issues:

1) What is exactly the panel for predicting COVID-19 mortality? What are the proteins in the panel and how the panel makes a prediction? What is the performance of this panel in terms of sensitivity, specificity, accuracy and ROC, in the patients studied here? How about the performance in independent patient cohorts/data sets?

See the more detailed description above; We included a new figure 5 where we explicitly show the proposed panels, why we choose them and how they perform at the three different time points in our cohort, as also depicted by ROC curves.

2) To identify the differentially expressed proteins, have the student t tests been corrected taking into account multiple testing? Have they set any threshold for fold-change?

The proteins reported as differentially regulated are selected based on their p-value and no additional filter has been set by using the fold change. In plasma proteomics of human samples, we and others find that fold changes are often modest and should therefore not be subjected to the same stringent fold change thresholds as one might otherwise apply. We are aware that the multiple-testing corrected p-values (q-values) reveal just a few proteins as differentially expressed in the data. We'd like to stress that our goal was not to present that individual proteins would predict patient outcome. Instead, we were looking for an optimal panel of protein expression values that can be used to predict mortality. We have chosen to analyze our data focusing on differential protein abundance at the earliest single timepoint. Indeed, we believe that in a clinical context, being able to

infer patient outcome based on a single early timepoint measurement is the most powerful tool. The plasma proteomics studies we compare our findings to centered primarily on longitudinal analysis, which require a different statistical setup and improve their significance values from observed trends (and therefore less sensitive to multiple hypothesis testing) thanks to multiple datapoints per protein across the time course. Despite this difference to our approach, we were very excited to observe that the from our data and analyses proposed protein panels discriminate well between survivors and non-survivors not only in our cohort but also in the data of the these two other studies performed independently by other researchers, with other study designs and other cohorts of patients from also different countries.

3) There are multiple serum proteomics datasets from COVID-19 patients in the literature, the authors mentioned some of them in the Discussion. A comprehensive, quantitative and comparative analysis of their data to the data sets from the literature should be performed to evaluate the value of the "panel", considering the sample size of this manuscript is too small.

See also comments made to the editor. We have taken this very seriously and requested the data from the Geyer (doi: <https://doi.org/10.1101/2021.02.22.21252236>) and Demichev studies (doi: <https://doi.org/10.1101/2020.11.09.20228015>). We like to thank these groups for making the data, from their MedRxiv studies, available to us. We have assessed the serum abundance of all proteins defined in our nine protein and sixteen protein panel in the Munich (Geyer) and Berlin (Demichev) cohorts, thereby taken time points that were close to our three time windows of serum collection, i.e. timepoint 1 close after submission to the ICU, time point 2 about 7 days after admission and timepoint 3 about 14 days after admission. We also filtered that data taking only severe state patients (WHO > 6). Although that meant that these cohort became of the same size as ours (or even smaller), we did observe that nearly all proteins from our defined 9 protein and 16 protein panel, at nearly all these timepoints showed similar trends as in our study. This we found extremely rewarding. In the Demichev study most of the proteins also showed significant p-values. In the Geyer study the trend was agreeable, but the statistics weaker. The data of all these analyses with protein abundances and p-values given for all proteins in the 9 and 16 protein panel are now included in the Supplementary figure 5.

Of note, our analysis strategy is very different from the analyses performed by Geyer and Demichev. While their analysis focused primarily on trends in protein abundance longitudinally, while patients were hospitalized, our analysis focused on differences in protein abundance between survivors and non-survivors at a single time point. Simply, in the Geyer and Demichev work a protein has a very good p-value if the longitudinal trend in abundance is statistically strong, in our work a protein has a good p-value when the protein abundance between survivors and non-survivors is substantial and reproducible in all patients. Reanalyzing the Geyer and Demichev data with our approach leads to somewhat smaller single time point p-values for their proteins, which still have very low (i.e. good) p-values when the trend in the concentration is taken.

Still the huge reproducibility in the trends seen for our sixteen-protein panel, as shown in the new supplementary figure 5 and 6, really indicates the strength of our panel, and moreover shows that our plasma proteome data are highly reproducible, even when the data is generated from other cohorts and by other researchers in different countries.

4) Figure 4: the concentrations of these proteins were estimated based on the literature with certain degree of accuracy (the overall R²=0.78), instead of direct measurement. It is misleading to directly use the absolute quant' 'mg/dL' as the y-axis.

It has not been our intention to potentially mislead the readers and we believe we have stated clearly that the given concentrations have been estimated using a calibration of "reported" plasma protein concentrations as described in Suppl Figure 1. In our experience readers/clinicians (also our collaborators from the clinic) do not appreciate LFQ values (MS based intensity parameters) and therefore we did opt to use these estimated mg/dL units, which are easier to understand. To address the reviewers point we now have added to the legend of Figure 4: "A. Serum abundance estimates, based on MS-based label free quantification (Suppl Figure 1), of the four type-3 cystatins,[...]".

Minor issues:

1) It is necessary to perform a more comprehensive literature review of the blood molecular (including protein) profiling of the COVID-19 patients in the Introduction.

We find this quite difficult to fully address. We have opted to cite what we believe to be the most relevant recent papers of good quality. It is true that a lot of blood molecular (including protein) profiling of the COVID-19 patients have appeared, and are appearing nearly every day, and we cannot include them all. We have now described also added the mentioned and cited the Galbraith paper and cite now half a dozen molecular profiling papers.

Additional studies we now cite in the revision are:

- *Shen et al, Proteomic and Metabolomic Characterization of COVID-19 Patient Sera, Cell 2020*
- *Park et al., In-depth blood proteome profiling analysis revealed distinct functional characteristics of plasma proteins between severe and non-severe COVID-19 patients, Sci Rep, 2020*
- *Shu et al., Plasma proteomics identify biomarkers and pathogenesis of COVID-19, Cell Immunity, 2020*

2) Typo: in some cases "COVID-19" was incorrectly spelt as "COVID19". So does "SARS-CoV-2".

This has now been corrected

3) Have the sera been de-activated to avoid potential infection? How much microliter sera was processed for proteomic analysis? How much peptides were injected into the MS?

The sera were not de-activated, and initial steps of RNA extraction were performed in a BSL-2 environment. All the sera were tested by PCR and only samples negative for SARS-CoV-2 were used in this study. Of note, in all the sera tested (larger cohort), only 2 samples were found to be weakly positive and those were thus excluded from the study.

For the question about the sample preparation, we have addressed this by making changes directly in the Methods section of the paper.

Referee #3 (Remarks for Author):

In this manuscript the authors have measured serum protein levels by mass spectrometry for samples collected from 33 individuals infected with SARS-CoV-2 (17 survivors, 16 deceased) at one to three timepoints following admission. On the order of ~450 proteins were quantified and several were found to be significantly different between patients that survived or not. The biomarkers of disease severity or mortality were then compared with biomarkers for COVID-19 severity identified in other studies (Geyer et al. 2021 and Demichev et al. 2020). To the authors own admission, this is a rather small cohort and there are important issues that make the evaluation of this result challenging even if one considers this work on its merit as a replication study. As it stands I find it difficult to be able to draw strong conclusions from the manuscript and also unsure about its value within the context of other related advances.

As stated in our paper, we believe that the significance of our paper, is partly coming from the confirmation of our findings with several of the same protein biomarkers predicting disease severity and or mortality as reported in two other manuscripts that appeared in parallel in MedRxiv. Our study was however quite differently focused, targeting a group of 33 people all with severe symptoms with the aim to find changes between survivors and non-survivors at the earliest time point. This is particularly interesting when it can help clinicians who have to make the call regarding which patients should be admitted to the ICU and what treatment they should be given. The studies by Demichev/Ralser and Geyer/Mann focused much more on longitudinal changes in serum protein abundances in a given patient, instead of directly comparing levels in between patients. In our revised manuscript we have taken their data and analyzed it following our approach. By selecting only severe case patients our number of COVID-19 deceased/survivors is on par with the cohorts in the studies performed by Demichev/Ralser and Geyer/Mann. The other studies indeed have more timepoints and more molecular markers, as does the study of Galbraith that next to plasma proteomics by mass-spectrometry and SOMAscan assays, performed cytokine profiling, and immune cell profiling via mass cytometry. With our resources and time, we honestly admit (also in the paper) that we cannot compete with these very large studies. Still, we show that the panel of markers we defined to predict disease severity/mortality can be used as well in the cohorts from Munich and Berlin, i.e. from a very different place in the world, obtained by other researchers and by using other approaches, without a priori knowledge about these other findings. This according to us justifies publication of the data, as it shows that plasma proteomics can reproducibly provide a panel of biomarkers, even when studies are performed in different laboratories and on different cohorts.

Major concerns

1 - I have concerns about the statistical analysis that were performed and the extent the authors can make the claims they do with the current dataset. There are two main issues, one has to do with the characterization of the cohort and potential confounders and the other more simply to do with multiple hypothesis testing.

1.1. - The authors tried to match to their best capacity the cohorts which is clear in regards to age and gender. There seems to be some difference in age that could already be important if they were to consider it in their statistical modelling. There may still be other aspects such as other underlying conditions that are not fully explained in the manuscript that may also need to be considered. In this regard, when comparing the two groups of patients and testing for the significant predictive value of a specific protein in distinguishing between them it is usually needed to consider such other factors (age, etc) as confounder. This would need to be achieved using a more sophisticated statistical test.

To clarify the (limited) influence of age and gender and their statistical interactions we have made and included a new analysis using a linear model. The ANOVA table of this analysis has been added in supplementary figure 2, which shows that neither age nor gender have a significant impact. The differences are caused by: (1) protein ID, i.e. proteins have different expression levels, (2) clinical outcome and (3) the interaction between protein and clinical outcome, therefore supporting the idea that protein levels can be used to infer outcome.

1.2 - In addition to description and incorporation of confounder into their statistical testing, the authors also need to account for multiple hypothesis testing (i.e. the number of tested proteins). The p-value cut-off used in the manuscript does not appear to take this into account. A stringent cut-off that considers 450 tests would put the p-value at 10^{-4} . Looking at the plots it would seem that there would be very few things that would be deemed significant. Considering the potential impact of accounting for the co-founders this may lead to no result being deemed significant.

Please see response to reviewer 1, under item 2.

2 - This work is focused on deriving protein biomarkers for COVID-19 mortality or severity. This is most related to two preprints (Geyer et al. 2021 and Demichev et al. 2020) and a recently published study (Galbraith et al. eLife 2021). While not strictly focused on comparing survivors from deceased these other works a larger in patient numbers and provide much of the value in terms of biomarker discovery and replication. While I certainly believe the topic to be of high importance and I am unsure about what this manuscript and work would add beyond what those already cover. Given the statistical issues from point 1, I have a difficult time in suggesting an application of the current data and manuscript that stands on its own and provides a significant advance in this area. If a predictive model was build on other larger datasets, then this cohort would perhaps serve as a useful independent test set but on its own it seems currently limited.

Please see the statement above, before 1. We find that showing in our work that plasma proteomics data can be reproduced in different cohorts, in different laboratories and with different operators and workflows is one of the main messages of our work. Moreover, the validity of the small panel of biomarkers, e.g. FETUB, KNG1, HRG, ITIH1, ITIH2, LCAT, HPR, SERPINA3 and AGT, as proposed by us and almost literally independently also by Demichev and Geyer, increases and may now be seriously further studied clinically

We hope that the editor and reviewer appreciate our reasoning

June 22, 2021

RE: Life Science Alliance Manuscript #LSA-2021-01099-TR

Prof. Albert J.R. Heck
Utrecht University
Biomolecular Mass Spectrometry and Proteomics Group
Padualaan 8
Utrecht 3584 CH
Netherlands

Dear Dr. Heck,

Thank you for submitting your revised manuscript entitled "Is there a serum proteome signature to predict mortality in severe COVID-19 patients?". We would be happy to publish your paper in Life Science Alliance pending final revisions necessary to meet our formatting guidelines.

- please re-phrase the title as a statement rather than a question
- please add a conflict of interest statement to your main manuscript text
- please consult our manuscript preparation guidelines <https://www.life-science-alliance.org/manuscript-prep> and make sure your manuscript sections are in the correct order
- please add your main, supplementary figure, and table legends to the main manuscript text after the references section
- please be sure to insert all Authors in the Authors Contribution section in your manuscript text
- please insert one figure per file if that is possible
- please use the [10 author names, et al.] format in your references (i.e. limit the author names to the first 10)
- please add callouts for Figures 1A, D, E; 2B, C; S2A, B; S3A, B; S5A; S6A, B to your main manuscript text

A. FINAL FILES:

B. MANUSCRIPT ORGANIZATION AND FORMATTING:

Sincerely,

Reviewer #1 (Comments to the Authors (Required)):

The authors have made a significant attempt to improve on the previous version of the manuscript. I still think their cohort is small with few significant changes surviving multiple test corrections. The biggest value added now is the comparison with the two other datasets which provide some reassurance that the markers they have identified can have some diagnostic value. Again they had to reduce the number of patients compared to try to make the analysis comparable and the AUCs or the specific pairwise comparison of protein levels is always consistent. This means of course that the other two studies also had few patients that either survived. I think this current study has value as an independent cohort characterised via plasma proteomics with promising indication of biomarkers that would need to be validated at larger scale for routine application. The data is available through a standard repository and the issue of cohort size is properly discussed.

I have no further concerns with this work.

June 24, 2021

RE: Life Science Alliance Manuscript #LSA-2021-01099-TRR

Prof. Albert J.R. Heck
Utrecht University
Biomolecular Mass Spectrometry and Proteomics Group
Padualaan 8
Utrecht 3584 CH
Netherlands

Dear Dr. Heck,

Thank you for submitting your Research Article entitled "A serum proteome signature to predict mortality in severe COVID-19 patients". It is a pleasure to let you know that your manuscript is now accepted for publication in Life Science Alliance. Congratulations on this interesting work.

DISTRIBUTION OF MATERIALS:

Again, congratulations on a very nice paper. I hope you found the review process to be constructive and are pleased with how the manuscript was handled editorially. We look forward to future exciting submissions from your lab.

Sincerely,
